# Barriers to effective communication among nurses and family members of patients admitted to the intensive care unit at Muhimbili National Hospital in Dar es Salaam: A descriptive qualitative study

**Menti Lastone Ndile**[1]*, **Scholastica Charles**[2], **Gift Lukumay**[3]

**1** Department of Clinical Nursing, School of Nursing, Muhimbili University of Health and Allied Sciences, Dar es Salaam, Tanzania, **2** Intensive Care Unit, Muhimbili National Hospital, Dar es Salaam, Tanzania, **3** Department of Community Nursing, School of Nursing, Muhimbili University of Health and Allied Sciences, Dar es Salaam, Tanzania

* menti.ndile@muhas.ac.tz

## Abstract

### Background

Effective communication in healthcare is essential for ensuring quality patient care. As healthcare shifts toward family-centered care, nurses are expected to engage families in information sharing and decision-making about their patients. However, barriers to effective communication in intensive care units (ICUs) exist. While global studies have addressed nurse-family communication, most are quantitative and conducted in contexts not similar to Tanzania. This study aimed to qualitatively explore the barriers nurses face in the Tanzanian context when communicating with family members in the ICUs at Muhimbili National Hospital, Dar es Salaam, Tanzania.

### Methods

A descriptive qualitative study was conducted using semi-structured interviews to explore communication barriers. Purposive sampling was used to select fifteen ICU nurses and twelve family members of ICU patients at Muhimbili National Hospital. All interviews were audio-recorded with consent and transcribed verbatim. Data were analyzed thematically following Braun and Clarke's six-phase framework.

### Results

The study identified two main themes emerging from the interviews: 1) Interpersonal barriers affecting the nurse-family relationship, and 2) Organizational barriers related to the work environment and resource limitations.

**Data availability statement:** All relevant data are within the paper and its Supporting Information files.

**Funding:** The author(s) received no specific funding for this work.

**Competing interests:** The authors have declared that no competing interests exist.

**Abbreviations:** ICU, Intensive Care Unit; MNH, Muhimbili National Hospital; MUHAS, Muhimbili University of Health and Allied Sciences; LMICs, Low and Middle Income Countries.

## Conclusion

The study highlights that improving communication in ICU settings goes beyond individual efforts; it requires investment in staff training, adequate infrastructure, and a supportive organizational culture. Addressing both interpersonal and organizational barriers through targeted interventions can foster trust, enhance family satisfaction, and contribute to improved patient outcomes in critical care environments.

## Background

Effective communication in healthcare is the clear, accurate, timely, respectful, and empathetic exchange of information between healthcare providers, patients, and families. It is essential for ensuring safe, patient-centered care and positive health outcomes [1]. It involves tailoring messages to align with the needs, values, and comprehension levels of diverse individuals while fostering trust, collaboration, and mutual understanding.

Nurses, due to their continuous bedside presence, play a pivotal role in bridging communication gaps between patients, families, and the multidisciplinary medical team. They are uniquely positioned to provide real-time updates, interpret clinical changes, offer emotional support, and advocate for patient and family needs. Recognizing that family in the care process enhances the quality of healthcare [1,2], healthcare systems are increasingly adopting family-centered care models, requiring nurses not only to deliver accurate medical information but also to actively engage families in discussions regarding treatment goals, prognoses, and care preferences [3].

Despite these expectations, effective communication in ICU settings is frequently challenged by a variety of barriers [4]. These include emotional stress, time constraints, and heavy workloads, among others. Such challenges can impede the establishment of trust, limit family involvement in decision-making, and negatively affect both patient care and family well-being.

Globally, many studies have researched nurse-family communication in ICU settings. However, the majority have employed quantitative approaches focusing on assessing communication outcomes [5–7], which may overlook the nuanced interpersonal, emotional, and contextual factors influencing communication. Furthermore, most existing qualitative research has been conducted in countries with contexts not similar to Tanzania, where healthcare systems, cultural expectations, and family dynamics differ from those in Tanzania. As a result, there is limited understanding of the specific barriers faced by nurses in Tanzanian ICUs when communicating with family members. Exploring these challenges through a qualitative lens is essential to developing contextually appropriate strategies that support effective communication and family-centered care. This study, therefore, aims to explore barriers to effective communication among nurses and family members of patients admitted to the ICUs at Muhimbili National Hospital in Dar es Salaam.

## Methods and materials

### Study design

This descriptive qualitative study used semi-structured interviews. The qualitative research design was selected to generate a thick description of the topic under study [8], ensuring a nuanced and context-rich exploration aligned with the study's objective of capturing the depth and complexity of communication in the healthcare setting. The study adhered to the Consolidated Criteria for Reporting Qualitative Research (COREQ) checklist to ensure methodological rigour and transparency. See Supporting Information S1 Text.

### Study setting

The study was conducted at Muhimbili National Hospital (MNH), Tanzania's national referral hospital, selected for its diverse ICU patient population. The hospital has two main ICUs: The medical and surgical ICUs, with a total bed capacity of around 30 and staffed with approximately 60 nurses. The nurses' educational backgrounds range from diploma-level to master's degrees in nursing. MNH's role as a major referral center for regional hospitals further reinforces its relevance as a study site, as it draws a heterogeneous mix of critically ill patients and their families. In Tanzania, nurse-to-patient ratios often exceed internationally recommended standards, a factor that may significantly impact the quality of nurse-patient communication.

### Participant recruitment and sample size

Purposive sampling, which refers to the deliberate choice of participants based on the qualities they possess [9] was employed to recruit two participant groups: (1) Nurses working in medical and surgical ICUs with ≥6 months of clinical experience, and (2) adult family members responsible for caring for their patients admitted to the ICUs for ≥72 hours. Family members were excluded if they had communication difficulties. The initial target sample size aimed for 18 participants per group with equal representation from both the medical and surgical ICUs. However, thematic saturation, the point at which no new themes emerged from consecutive interviews, was reached earlier during iterative data collection and analysis. Consequently, the final sample consisted of 15 nurses and 12 family members.

### Data collection and procedure

The study was conducted from June 5th to July 10th, 2023. Semistructured interview guides tailored separately for nurses and family members were developed through a comprehensive literature review on communication barriers in critical care settings. To ensure contextual relevance, the guides were pretested with two nurses and two family members (excluded from the final sample) and refined for clarity, eliminating ambiguities in phrasing. The final guides featured four open-ended core questions aligned across both groups (See Supporting Information S2 Text). One example of the question was; *"Describe your experiences communicating with families/nurses during ICU admissions",* paired with follow-up probes to elicit deeper reflections (*"Can you elaborate on what made that interaction challenging?"*).

Nurse managers from medical and surgical ICUs assisted in identifying eligible nurse participants, while family members were recruited through patient registers. Participants were provided with the information about the purpose of the study, the modality of data collection, confidentiality safeguards and the right to withdraw. Verbal consent to participate in the study was obtained, followed by written consent on the interview day. Interviews were conducted privately in a designated hospital room during weekdays to minimize interruptions and protect confidentiality. To optimize participant comfort and data richness, all sessions were conducted in Kiswahili, Tanzania's national language, by a bilingual researcher fluent in both Kiswahili and English. Interviews lasted 60–90 minutes, allowing sufficient time for nuanced discussions. Audio recordings were made with the participant's consent, supplemented by field notes that captured non-verbal cues.

### Data analysis

The audio-recorded interviews were transcribed verbatim. An inductive thematic framework by Braun and Clarke's six-step framework [10] was employed to identify patterns and themes directly from the data. The process started with authors familiarizing with data through repeated reading of transcripts, followed by identifying and labelling meaningful text segments with descriptive codes in Kiswahili to preserve original meaning. These codes were subsequently translated into English for collaborative analysis, with translation accuracy verified by bilingual team members to mitigate misinterpretation. Codes with similar patterns were iteratively grouped and labelled with tentative themes and subthemes. This phase involved rigorous discussion among three co-authors (SC, MN, GL) and an external qualitative expert to ensure analytical consistency and reduce researcher bias. Tentative themes were critically re-examined against the original transcripts and coded data. Through constant comparison, themes were revised or redefined to enhance coherence and ensure they authentically represented the dataset. The overall data analysis produced two primary themes and seven subthemes supported by representative participant quotes for each theme.

### Trustworthness

Data collection and analysis were conducted concurrently by all authors, using constant comparison of emerging themes across transcripts. This iterative approach allowed for refinement of the interview guide and supported the achievement of theoretical saturation [11]. The analytical process was transparently documented, including audio and raw transcripts data, coding procedures, and theme development, aiming to illustrate the analytical pathway. To enhance the credibility of the analysis and mitigate researcher bias, an independent qualitative research expert from Muhimbili University of Health and Allied Sciences reviewed the coding framework and theme generation. Additionally, preliminary findings were shared with some participants to get their thoughts on how the analysed data reflected their views. Detailed contextual information about the ICU setting and participant demographics was provided in the study to support the transferability of findings to similar settings.

### Ethical approval and consent to participate

The study received ethical approval from the Institutional Review Board of Muhimbili University of Health and Allied Sciences (MUHAS) (Ref. No: DA.282/298/01.C/1705), with additional permission granted by Muhimbili National Hospital management (Ref. No: MNH/CRTCU/Perm/2023/385). Before starting interviews, the principal researcher explained to participants the study's purpose and measures to safeguard confidentiality. Furthermore, they were informed that their anonymized data would be shared for research purposes and the public good. Participants were assured of their right to decline participation or skip sensitive questions at any time. Written informed consent was obtained from participants before an interview, ensuring voluntary and transparent engagement of the participants throughout the research process.

## Results

### Participant socio-demographic characteristics

A total of 15 nurses and 12 family members of patients admitted to the ICU were interviewed. Among the participants, nine were males and 18 were females. The majority, participants (n = 26; 96.3%) were aged between 30 and 49 years. Additionally, 22 participants (81%) had completed secondary education or higher (Table 1).

### Emerged themes and sub-themes

Through thematic analysis, seven sub-themes emerged from coded data by identifying patterns of similarity and divergence. These sub-themes were subsequently consolidated into two overarching themes.: *Interpersonal Barriers* and *Organizational Barriers*. The themes and subthemes are presented in Table 2.

**Table 1. Participants characteristics.**

| | | Family member s(n = 12) | Nurses (n = 15) |
|---|---|---|---|
| Gender | Male | 3 | 6 |
| | Female | 9 | 9 |
| Age | 30–39 | 7 | 10 |
| | 40–49 | 4 | 5 |
| | 50–59 | 1 | 0 |
| Level of education | Primary education | 5 | 0 |
| | Secondary education | 2 | 0 |
| | College and above | 5 | 15 |
| Years of working experience | Less than 5 years | | 6 |
| | More than 5 years | | 9 |
| Duration of caregiving | Less than 2 weeks | 8 | |
| | More than two weeks | 4 | |
| Relationship to the patient admitted in ICU | Parent | 5 | |
| | spouse | 4 | |
| | Sibling | 1 | |
| | Others | 2 | |
| Employment status | Formal employment | 4 | |
| | Self-employed | 8 | |

**Table 2. Themes and sub-themes.**

| Theme | Subtheme | Codes |
|---|---|---|
| **Interpersonal barriers** | Nurses attitude | • Indifference toward families<br>• Lack of empathy |
| | Inadequate Communication Skills | • Failure to actively listen<br>• Dismissive interactions with families<br>• Difficulty expressing complex information<br>• Inconsistent communication styles among nurses |
| | Preconceived Beliefs and Mistrust towards Nurses | • Perception that nurses are rude<br>• Perception that nurses are unfriendly |
| **Organizational barriers** | Staff workload | • Overwhelmed by patient care demands<br>• Nurse-to-patient ratio high<br>• Burnout impacting nurse-family interaction |
| | Short visiting hours | • Restrictive visiting hours<br>• Limited time for family-patient engagement<br>• Inflexible visitation policies |
| | Lack of counselling rooms | • Lack of private space for sensitive conversations<br>• Lack of confidentiality in shared spaces |
| | Role-Specific Communication Restrictions | • No standardized communication protocols<br>• Absence of structured updates for families<br>• Unclear communication roles |

## Theme 1: Interpersonal barriers

This theme highlights how nurse-family interactions significantly impact family members' trust, emotional well-being, and perceptions of care quality. The theme consists of three subthemes: 1) Nurses' attitude, 2) Inadequate communication skills and comprehension and 3) Preconceived beliefs and mistrust towards nurses

**Nurses' attitude.** Family members reported feeling invalidated when requesting updates about their loved ones. One participant recounted frustration at being met with suspicion rather than collaboration:

*"Sometimes, you want to ask a nurse for information about your patient, but instead, you're asked, 'Why do you want to know all that? Just see your patient and go." (Relative 12)*

Instances of harsh reprimands instead of compassionate guidance exacerbated distress. A family member described feeling humiliated after an unintentional error:

*"Some of us didn't know hospital rules. I touched my patient without sanitizing, and the nurse shouted at me instead of explaining kindly." (Relative 6)*

Negative encounters heightened anxieties about how voiceless patients might be treated. One family member stated:

*"If I'm treated poorly, I go home wondering… what happens to my patient who can't speak for himself?" (Relative 3)*

**Inadequate communication skills and comprehension.** Nurses reported greater ease communicating with educated family members, indicating disparities in health literacy. One nurse explained:

*"It's easier to talk with educated family members; they understand you. With others, you have to simplify everything to get understood." (Nurse 8)*

There were challenges in message relay. A nurse described how details often became distorted as they passed through family members:

*"It happens that after sharing some information with a representative about their patients, when they go to share with the rest, they say something different…" (Nurse 2)*

**Preconceived beliefs and mistrust towards nurses.** Nurses observed that preconceived mistrust from family members hindered good communication and relationships. One nurse highlighted how these assumptions strained interactions:

*"When family members arrive at the hospital already convinced that nurses are harsh or indifferent to their concerns, it creates an immediate barrier to establishing trust and good cooperation."(Nurse 4)*

### Theme 2: Organizational barriers

This theme explores how heavy workloads, limited visiting hours, lack of private spaces, and communication restrictions hinder effective nurse-family interactions. The subthemes are presented and supported by quotes.

**Staff workload.** In high-stakes environments like ICUs, nurses reported being stretched. One nurse described the tension between clinical duties and family interactions:

*"Patients admitted to the ICU are very sick and need close observation. You become so occupied with the care to the point you don't have time to talk much with patient family members." (Nurse 7)*

This strenuous situation was compounded by understaffing. As another nurse explained:

*"You may find yourself caring for two critically ill patients. In that kind of situation, it becomes difficult to have meaningful conversations with family members about their patients." (Nurse 10)*

**Short visiting hours.** Limited visiting time reduced the opportunity for family members to receive updates from nurses. One family member emphasized the inadequacy of time for both connection and updates:

*"We are allowed to visit our patients just for a while, that's not even enough for nurses themselves to share information with us." (Relative 2)*

**Lack of a counselling room.** Nurses often had to provide sensitive information in non-private environments, risking confidentiality. One nurse described the situation:

*"We normally provide patient information to family members at the bedside. If it's something very confidential, then we ask for space in the in-charge's office for privacy or the doctor's room."* (Nurse 3)

Family members felt uncomfortable asking personal questions in shared spaces due to a lack of privacy, as explained by one of them.

*"You may want to ask the nurse sensitive information about your patient, but you find a lot of other family members with their patients… so you refrain… one time I asked the nurse, she spoke loudly… everyone around could hear..."* (Relative 9)

**Role-specific communication restrictions.** Nurses expressed limitations in the information they were authorized to share, leading to frustration for family members. One nurse explained:

*"I can't share everything, some details only doctors disclose. But families keep asking us, and we're stuck saying, 'Ask the doctor,' which feels dismissive."* (Nurse 1)

Family members were often redirected without guidance, feeling confused about whom to approach, as narrated by one of them:

*"You ask the nurse about your patient, and you get a response like 'go ask the doctor,' but I always see the nurse with my patient. When you tell me to ask the doctor… where?" (Relative 4)*

## Discussion

This study explored barriers to effective communication between nurses and family members of critically ill patients in a Tanzanian ICU. Two overarching themes emerged: interpersonal barriers and organizational barriers. These findings reveal a complex interplay between individual, relational, and organizational factors that hinder communication and, consequently, impact care quality and family satisfaction in the context of family-centered care.

Participants described how individual nurse behaviour shaped their emotional experience and trust in the healthcare team. Negative interactions led to feelings of alienation and anxiety, reinforcing the critical role of empathy and relational care in hospital settings [12–14]. Conversely, positive interactions with kind and informative nurses were deeply valued [7] but inconsistently experienced, pointing to variability in interpersonal competence[15]. Findings from other previous studies align with our study, that competency in effective communication is inadequate [16–20]. This suggests the need for in-house refresher courses or mentorship programs to strengthen empathetic communication and customer care suited for high-pressure environments like the ICU. These initiatives can support nurses in developing and sustaining the required skills.

Communication breakdowns were also attributed to characteristics and perceptions held by family members. Preconceived mistrust and assumptions about nurses being unfriendly or unhelpful align with previous literature suggesting that patient families often approach healthcare encounters with anxiety, bias, or defensive attitudes [4]. These

initial attitudes can obstruct rapport-building and deepen misunderstandings. Additionally, disparities in health literacy emerged as a barrier to effective dialogue. Family members with low literacy struggled to comprehend medical explanations or instructions from nurses. The finding relates to other previous studies where low health literacy is associated with the risk of miscommunication, reduced comprehension about medical information among family members [21,22]. The finding emphasizes the need to empower nurses to be able to cater for diverse family members when communicating health messages.

Staffing shortages and heavy workloads were also identified as key organizational impediments to communication. With limited nurse-to-patient ratios, especially in critical care, nurses were often unable to allocate sufficient time to interact meaningfully with each patient's family members. This resonates with studies from other LMICs where nurse burnout and workload have been associated with reduced quality of patient and family engagement [23–25], therefore leading to lower satisfaction with healthcare services. Addressing staffing shortages and redistributing tasks could free nurses' time for meaningful family interactions.

Infrastructural limitations further constrained effective communication. The lack of private spaces for sensitive conversations led to confidentiality breaches and discomfort for staff and families. The practice of using ad hoc spaces, such as doctors' rooms or in-charge offices, reflects broader infrastructural inadequacies in critical care units, especially in resource-limited settings [4]. Efforts should be made to designate private areas for confidential family discussions within ICUs that may include low-cost modifications such as a partitioned consultation area or scheduled use of shared offices, enhancing both the dignity and effectiveness of communication.

Organizational policies regarding communication roles and visiting hours also affected interactions. Role ambiguity between nurses and doctors often left family members confused about whom to approach for information. This dissonance highlights the need for clearer protocols that define the roles and responsibilities of nurses and doctors in providing patient updates. While in our study, family members acknowledged that nurses and indispensable in patient information communication, a study conducted in Jordan showed that physicians were highly regarded and sought more than nurses [16]. Strengthening multidisciplinary family meeting practices can help standardize information-sharing and reduce confusion [26].

Visiting time policies in the ICU differ [27].The majority of ICUs apply some restrictive visitation policies, as in Tanzania, which reduce opportunities for timely and comprehensive interaction and information sharing between nurses and family members [18]. Flexible visitation policies in ICUs have been shown to significantly reduce anxiety and depressive symptoms among family members and healthcare providers by fostering a sense of involvement and transparency in patient care [28,29]. However, implementing a flexible visitation policy in low-resource countries would require substantial investments in healthcare infrastructure, human resources, and overall health system strengthening. Such a policy may also present challenges for ICU nurses, including increased psychological distress due to heavier workloads, disruptions to care routines, and emotional strain from managing complex family issues.

## Implications

These findings have several important implications for practice. First, there is a clear need for targeted communication training for nurses that incorporates strategies for engaging with families empathetically.with attention to navigating health literacy differences.

Second, hospitals should invest in organizational improvements, including the establishment of structured communication protocols between nurses and family members. Such measures can safeguard confidentiality, reduce misinformation, and ensure that family members feel heard and respected.

Third, staffing policies need to be reviewed to alleviate nurse workload, particularly in high-dependency units. Adequate staffing not only improves patient outcomes but also provides nurses with the space to attend to family concerns more comprehensively.

## Strengths and limitations

This may be the first study to explore communication barriers between nurses and family members of patients admitted to an ICU in Tanzania, providing an in-depth understanding of the situation. However, several limitations should be noted. First, the experiences of nurses and family members were taken from a single hospital, which may limit the generalizability of the findings to other settings. Caution should therefore be exercised when applying these results elsewhere. Nonetheless, the study offers rich contextual information that enhances the relevance and transferability of the findings. Another limitation is that the study was conducted from the perspective of the nursing discipline. As a result, the findings primarily contribute to enhancing nursing care practices in the ICU. However, communication between healthcare providers and family members is inherently interdisciplinary and requires collaborative input from all relevant healthcare professionals. The information provided by the participants may have been influenced by social desirability bias, as the participants might have felt inclined to give responses that impress the interviewer. However, the researcher emphasized the importance of honest responses to ensure the collection of meaningful data that could contribute to improving patient care. The final important limitation worth mentioning is that the majority of the participants were female, and therefore, their views may have been influenced by gender related factors.

## Conclusion

Effective communication between nurses and family members is central to quality care in the ICU. This study demonstrates that enhancing communication requires more than individual effort; it demands system investment in training, infrastructure, and organizational culture. Addressing these barriers through targeted interventions can strengthen trust, improve family satisfaction, and ultimately lead to better patient outcomes in critical care settings.

## Supporting information

**S1 Text. Consolidated criteria for reporting qualitative research (COREQ) checklist.**
(DOCX)

**S2 Text. Interview guide.**
(DOCX)

**S3 Text. Excerpts from transcripts.**
(DOCX)

## Acknowledgments

The authors would like to thank all study participants and the Muhimbili National Hospital management for the support in facilitating data collection.

## Author contributions

**Conceptualization:** Menti Lastone Ndile, Scholastica Charles.

**Data curation:** Menti Lastone Ndile, Scholastica Charles.

**Methodology:** Menti Lastone Ndile, Scholastica Charles, Gift Lukumay.

**Writing – original draft:** Menti Lastone Ndile, Scholastica Charles, Gift Lukumay.

**Writing – review & editing:** Menti Lastone Ndile, Scholastica Charles, Gift Lukumay.

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
