## [Decision Letter · Decision Letter 0]

9 Jul 2025

PONE-D-25-29279Barriers to effective communication among nurses and family members of patients admitted to the intensive care unit at Muhimbili National Hospital in Dar es Salaam: A descriptive qualitative studyPLOS ONE

Dear Dr. Ndile,

Thank you for submitting your manuscript to PLOS ONE. After careful consideration, we feel that it has merit but does not fully meet PLOS ONE’s publication criteria as it currently stands. Therefore, we invite you to submit a revised version of the manuscript that addresses the points raised during the review process.

We look forward to receiving your revised manuscript.

Kind regards,

Fatma Refaat Ahmed, Ph.D.

Academic Editor

PLOS ONE

Journal Requirements:

Reviewers' comments:

Reviewer's Responses to Questions

**Comments to the Author**

1. Is the manuscript technically sound, and do the data support the conclusions?

Reviewer #1: Yes

Reviewer #2: Yes

2. Has the statistical analysis been performed appropriately and rigorously? 

Reviewer #1: N/A

Reviewer #2: Yes

3. Have the authors made all data underlying the findings in their manuscript fully available?

Reviewer #1: Yes

Reviewer #2: Yes

4. Is the manuscript presented in an intelligible fashion and written in standard English?

Reviewer #1: Yes

Reviewer #2: Yes

5. Review Comments to the Author

Reviewer #1: The authors followed the scientific equiry process for qualitative studies. The following are my comments:

Thank you for the opportunity to review this paper titled “Barriers to effective communication among nurses and family members of patients admitted to the intensive care unit at Muhimbili National Hospital in Dar es Salaam: A descriptive qualitative study.” This paper has merits as it provides insights into improving communication between ICU nurses and family members of patients. The paper is generally well-written, and the following comments can improve this paper.

General comment

There many typographical errors that need attention. Authors should inspect all sentences

Abstract

• Well-written and clear

Introduction

• Line 48, correct the typographical error

• Facts are presented in line 49 to 52, but there is no citation provided

• The authors are reverting to the use of long-form and short form of the term ‘intensive care unit’, The principle of use of abbreviation needs to be followed for consistency

Methods

• There are many ICUs at MNH. However, information is lacking regarding their description in the study setting and which ICUs were included and why, and how the participants were selected across the ICUs.

• In line 103, ICU charge nurses should be revised, it is not clear

• There is a typo of full stop in line 110

• The sentence from line 114 to 116 needs revision

• Line 143 has a bracket typo

Results

• Male and female in line 152 should be in plural

• A sentence in line 152 - 153 needs to be revised to read “The majority, participants (n=26; 96.3%), were aged between 30 and 49 years”

• In line 157, information regarding the initial analysis that yielded 52 codes should be omitted. Authors should only present the results and not the process of analysis

• Authors should only present findings and avoid discussion. Statements such as in line 185 to 186 (These insights underscore how individual nurse behavior can deeply influence family members’ trust, emotional state, and perception of care quality) should be avoided in the findings section. They should be part of the discussion.

Discussion

• Citations in line 255 – 256 need to be merged

• Sentences from line 297 – 301 need to be revised for clarity

• Authors may expand on limitations to include social desirability

Conclusion

Well-summarized

Reviewer #2: Hello. Here are some issues from my observation

1. In astract, line no 21-22, as this is the report, I think it should read "This atudy aimed at..."

2. The backgroud is too long, I suppose you rephrase it.

3. Line no 86, there is no need of citing (no idea what was cited here). If need be, the citation ahould be made in line 90-91(definition of saturation)

4. There is a bracket hanging in line 143 (a single bracket)

5. In table 1, since your sampling was purposive, why could'nt you balance the gender between participants?

6. Since there is no any participant below the age of 30, is it necessary for this category to be included?

7. In line 254-255, I suppose all the four citation stay in the same bracket, e.g. [15-19] instead of [15, 16] [17-19]

That is all from my observation. Thank you and all the best

6. PLOS authors have the option to publish the peer review history of their article (what does this mean? ). If published, this will include your full peer review and any attached files.

**Do you want your identity to be public for this peer review?** For information about this choice, including consent withdrawal, please see our Privacy Policy .

Reviewer #1: No

Reviewer #2: No

---

## [Author Response · Author response to Decision Letter 1]

17 Jul 2025

Dear editor and reviewers.

I appreciate your comments about the manuscript. I have responded to all comments (see attached Response to Reviewer sheet), and I'm sure your comments have improved the manuscripts significantly. I also confirm that the participants consented that the data in de-identified form can be shared including published for public good.

Thank you

---

## [Decision Letter · Decision Letter 1]

31 Jul 2025

Barriers to effective communication among nurses and family members of patients admitted to the intensive care unit at Muhimbili National Hospital in Dar es Salaam: A descriptive qualitative study

PONE-D-25-29279R1

Dear Dr. Ndile,

We’re pleased to inform you that your manuscript has been judged scientifically suitable for publication and will be formally accepted for publication once it meets all outstanding technical requirements.

Kind regards,

Fatma Refaat Ahmed, Ph.D.

Academic Editor

PLOS ONE

Additional Editor Comments (optional):

Reviewers' comments:

Reviewer's Responses to Questions

**Comments to the Author**

1. If the authors have adequately addressed your comments raised in a previous round of review and you feel that this manuscript is now acceptable for publication, you may indicate that here to bypass the “Comments to the Author” section, enter your conflict of interest statement in the “Confidential to Editor” section, and submit your "Accept" recommendation.

Reviewer #1: All comments have been addressed

2. Is the manuscript technically sound, and do the data support the conclusions?

Reviewer #1: Yes

3. Has the statistical analysis been performed appropriately and rigorously? 

Reviewer #1: N/A

4. Have the authors made all data underlying the findings in their manuscript fully available?

Reviewer #1: Yes

5. Is the manuscript presented in an intelligible fashion and written in standard English?

Reviewer #1: Yes

6. Review Comments to the Author

Reviewer #1: The authors have addressed all the comments and I do not have any more comments. I think you can proceed with further steps.

7. PLOS authors have the option to publish the peer review history of their article (what does this mean? ). If published, this will include your full peer review and any attached files.

**Do you want your identity to be public for this peer review?** For information about this choice, including consent withdrawal, please see our Privacy Policy .

Reviewer #1: **Yes: ** Dr. Joel Seme Ambikile

---

## [Editor Report · Acceptance letter]

PONE-D-25-29279R1

PLOS ONE

Dear Dr. Ndile,

I'm pleased to inform you that your manuscript has been deemed suitable for publication in PLOS ONE. Congratulations! Your manuscript is now being handed over to our production team.

Kind regards,

on behalf of

Dr. Fatma Refaat Ahmed

Academic Editor

PLOS ONE